# Synthesis and Evaluation of Biological Activities of Bis(spiropyrazolone)cyclopropanes: A Potential Application against Leishmaniasis

**DOI:** 10.3390/molecules26164960

**Published:** 2021-08-17

**Authors:** Olalla Barreiro-Costa, Gabriela Morales-Noboa, Patricio Rojas-Silva, Eliana Lara-Barba, Javier Santamaría-Aguirre, Natalia Bailón-Moscoso, Juan Carlos Romero-Benavides, Ana Herrera, Cristina Cueva, Lenin Ron-Garrido, Ana Poveda, Jorge Heredia-Moya

**Affiliations:** 1Centro de Investigación Biomédica (CENBIO), Facultad de Ciencias de la Salud Eugenio Espejo, Universidad UTE, Quito 170527, Ecuador; olallabarreirocosta@gmail.com (O.B.-C.); projas1@usfq.edu.ec (P.R.-S.); 2DNA Replication and Genome Instability Unit, Grupo de Investigación en Biodiversidad, Zoonosis y Salud Pública (GIBCIZ), Instituto de Investigación en Zoonosis-CIZ, Facultad de Ciencias Químicas, Facultad de Ciencias Agrícolas, Universidad Central del Ecuador, Quito 170521, Ecuador; gemoralesnoboa@hotmail.com (G.M.-N.); eliluci1@hotmail.com (E.L.-B.); jrsantamaria@uce.edu.ec (J.S.-A.); ljron@uce.edu.ec (L.R.-G.); 3Departamento de Ciencias de la Salud, Universidad Técnica Particular de Loja, Loja 1101608, Ecuador; ncbailon@utpl.edu.ec (N.B.-M.); agherrera5@utpl.edu.ec (A.H.); mccueva5@utpl.edu.ec (C.C.); 4Departamento de Química, Universidad Técnica Particular de Loja, Loja 1101608, Ecuador; jcromerob@utpl.edu.ec

**Keywords:** bis(spiropyrazolone)cyclopropanes, drugs, leishmaniasis cytotoxicity, ADME

## Abstract

This work focuses on the search and development of drugs that may become new alternatives to the commercial drugs currently available for treatment of leishmaniasis. We have designed and synthesized 12 derivatives of bis(spiropyrazolone)cyclopropanes. We then characterized their potential application in therapeutic use. For this, the in vitro biological activities against three eukaryotic models—*S. cerevisiae*, five cancer cell lines, and the parasite *L. mexicana*—were evaluated. In addition, cytotoxicity against non-cancerous mammalian cells has been evaluated and other properties of interest have been characterized, such as genotoxicity, antioxidant properties and, in silico predictive adsorption, distribution, metabolism, and excretion (ADME). The results that we present here represent a first screening, indicating two derivatives of bis(spiropyrazolone)cyclopropanes as good candidates for the treatment of leishmaniasis. They have good specificity against parasites with respect to mammalian cells.

## 1. Introduction

Heterocycles are common structural motifs in marketed drugs. Many have at least one heterocycle in their structure and are the target of medicinal chemistry in the drug discovery process [1]. Nitrogen-containing rings in particular play an important role in drug development due to their wide variety of therapeutic and pharmacological properties. Pyrazoles and their derivatives can be analgesic, anti-inflammatory, antipyretic, antioxidant, anticonvulsant, antidepressant, antihyperglycemic, antimicrobial, antiviral, antitumor, hepatoprotective, and spasmolytic [2,3,4]; several drugs currently on the market have a pyrazole ring as the key structural motif [5].

Derivatives of 2,4-dihydro-3*H*-pyrazol-3-one include edaravone (**1**), a powerful antioxidant [6], and 4,4′-(arylmethylene)bis(1-phenyl-3-methyl-1*H*-pyrazol-5-ol) (**2**). These have a wide spectrum of reported biological activities and have been used as anti-inflammatory, antipyretic agents, stimulants of gastric secretions, antidepressants, antibacterial agents and filaricides. These pyrazole-5-ols have been used as fungicides, pesticides, insecticides, and dyes [7]. However, despite the great variety of reported activities of these heterocycles, there are virtually no studies on leishmanicidal [8] or cytotoxic activity against tumor cells. Of the few similar compounds studied, the derivatives of phenyl pyrazolones are interesting despite not showing good leishmanicidal activity because they are intermediates in the synthesis of 4,4′-(arylmethylene)bis(1-phenyl-3-methyl-1*H*-pyrazol-5-ols) [9]. An analysis of these compounds shows that their structure is similar to 3,3′-(arylmethylene)bis(2-hydroxynaphthalen-1,4-diones) [10] and 3,3′-diindolylmethanes [11], which possess potent leishmanicidal activity. Thus, the study of 1*H*-pyrazole-5-ol and similar compounds against these parasites is of interest.

Thus, our laboratory has worked on the synthesis of 4,4′-(arylmethylene)bis(1-phenyl-3-methyl-1*H*-pyrazole-5-ols) and assayed them against in vitro models previously implemented in our laboratory. We found that these compounds have good antioxidant activity in vitro, as well as cytotoxicity against the RKO cell line [12]. We found good leishmanicidal activity against promastigotes of *Leishmania mexicana* and the evaluation of the anti-trypanosomatid activity of these compounds is in progress and will be reported elsewhere.

A structural analysis of the active principles of medicinal agents that are currently on the market shows that several of them incorporate cyclopropane. Some of these drugs have a high market demand including montelukast and ciprofloxacin. Molecules that incorporate this ring in their structure have received considerable attention because they exhibit a wide range of biological activities, including enzymatic inhibition, insecticide, analgesic, antifungal, phytotoxic, antibiotic, antifungal, antiviral, antitumor and hormonal activities [13,14,15,16,17,18].

A diverse range of biological activities have also been reported for spirocyclopropanes, such as anti-inflammatory, analgesic, anticancer, and cytotoxic activity [19,20,21,22]. The attention from compounds that incorporate this structure has not been limited to molecules of synthetic origin, but rather have focused on natural products because this functionality has been reported in several naturally occurring compounds [23,24,25,26]. In the case of spirocyclopropanes bound to heterocycles there are few reports of biological activity. Those that can be found are limited to documenting the antiviral activity [27,28], and their activity as inhibitors of spliceosome [29], alpha-l-fucosidase [30], β-lactamase [31] and non-nucleoside inhibitors of HIV-1 reverse transcriptase [32]. Of note, ledipasvir is used to treat hepatitis C and has this structural element [33].

Spirocyclopropanes can be easily synthesized from 4,4′-(arylmethylene)bis(1-phenyl-3-methyl-1*H*-pyrazol-5-ol) [34] or directly from 3-methyl-1-phenyl-2-pyrazolin-5-one and benzaldehydes **3** in one step [35]. However, their bioactivity is largely uncharacterized, and the only report found is focused in the inhibitory activity of advanced glycation products (PGA) for the treatment of schizophrenia [36].

Therefore, continuing with our study on the bioactivity of heterocyclic compounds [37,38] and in view of the great structural similarity that spirocyclopropanes present versus the previously studied bispyrazole-5-ol and the antecedents discussed, we report here the synthesis of a series of bis(spiro-2,4-dihydro-3*H*-pyrazol-3-one) cyclopropanes **4**, starting from 4,4′-(arylmethylene)bis(1-phenyl-3-methyl-1*H*-pyrazol-5-ol) **2**. Their biological activities were evaluated in three models: yeast *Saccharomyces cerevisiae*, promastigotes of *Leishmania mexicana*, five human tumor cell lines, and one normal cell line. Additional tests were performed to decipher its mechanism of action (antioxidant and genotoxicity).

## 2. Results

### 2.1. Synthesis

The synthesis of a small library of 4,4′-(arylmethylene)bis(1*H*-pyrazol-5-ols) **2a-u** was performed through an efficient synthetic approach using NaOAc as catalyst as previously reported (Scheme 1) [12]. The optimal conditions consist of the reaction of the pseudo-three-component substituted benzaldehyde **3a-u** with two equivalents of 3-methyl-1-phenyl-2-pyrazolin-5-one (**1**) at room temperature using 70% EtOH as solvent. This leads to the corresponding 4,4′-(arylmethylene)bis(1-phenyl-3-methyl-1*H*-pyrazol-5-ol) derivatives **2a-u** in good to excellent yield (Table 1).

The spectroscopic data and melting points of compounds previously reported agreed with literature values. As expected, the reaction time showed the same substituent effect as previously observed [12], except for **2g**. These had strong electron-donating dimethylamino group in **3g,** and the possibility of the formation of a tautomeric quinoid structure [39,40] could explain the reduction of the activity of this benzaldehyde (entry 8).

The synthesis of **4** was performed using the electrocatalytic cyclization of **2** as previously reported (Scheme 1) [34]. The reaction was made using a 6 V, 800 mA D.C. power supply, and pure product was obtained after filtration in good to moderate yields (Table 1). However, no product was obtained when the starting bispyrazole has hydroxyl groups in its structure. A complex mixture was observed in these cases.

The presence of many products in these mixtures suggests the possibility of other reactions that compete with the desired one. Spirocyclopropanes **4** are known to isomerize to 4-[(pyrazol-4-yl)-methylene]pyrazolones when in solution, especially in DMSO and high temperature [41]. In our experience, however, this isomerization is observed even at low temperature. Structure **2** is reported to be thermo labile and suffer a retro-Michael reaction when in solid state or in solution; this stability is influenced by the presence of the different substituents in the aromatic ring [42]. Electron-acceptor groups increase the stability of **2**, while electron-donor groups reduce their stability. Thus, a more complex mixture is expected. In the case of **2g**, the main product of the reaction is the *1H*-pyrazol-5(4*H*)-one **5** with a yield of 60% (Entry 7). For **2l**, it was possible to isolate a small amount of bis(spiropyrazolone)cyclopropane **6**, which was characterized by 2D-NMR. This also suggests that, in addition to the retro-Michael reaction, there would also be decomposition reactions of the bis(spiropyrazolone)cyclopropanes formed.

### 2.2. Biological Activity of Bis(spiro pyrazolone)cyclopropanes against Yeast S. cerevisiae, and Human Cancer Cell Lines

The sensitivity to all synthesized bis(spiropyrazolone)cyclopropanes **4** was evaluated against five human cancer cell lines (RKO, A-549, MCF-7, PC-3 and HeLa) and budding yeast *S. cerevisiae* (W303 strain). First, the effect on viability of human cancer cell lines was assessed using the MTS assay. Cells were treated with 100 µM of each derivative for 48 h and doxycycline was used as control. The strongest effect was obtained with the derivative **4s**, with 70.23% of cellular inhibition in the RKO cell line (Appendix A).

Next, we determine the inhibitory concentration fifty %, IC_50_ (Table 2). The lines most sensitive to the compounds (RKO, PC-3 and Hela) were selected. The IC_50_ was established in those that inhibited more than 50% of cell growth in any of these three cell lines (Appendix A). Here, cells were exposed to each compound in concentrations ranging between 15 and 150 µM. The IC_50_ calculated for the derivatives shows values from 49.79 µM to 113.70 µM. Derivative **4r** was the most potent in RKO, PC-3 and HeLa cell lines with IC_50_ values of 60.70 µM, 49.79 µM, and 78.72 µM, respectively. The less potent derivatives were **4d** and **4q**. Finally, the cell morphology of RKO, PC-3 and HeLa cell lines exposed to the IC_50_ of the most active bis(spiro-2,4-dihydro-3*H*-pyrazol-3-one) cyclopropanes for 48 h was checked microscopically to confirm the effect of the compounds (Appendix A). Overall, our results show that derivatives **4d**, **4k**, and **4q-s** exert a reliable inhibitory activity on human cancer cell lines RKO, PC-3 and HeLa, but not on A-549 nor MCF-7. However, the IC_50_ values are relatively high for potential anticancer drugs.

Next, we determined the IC_50_ of bis(spiropyrazolone)cyclopropanes on a different eukaryotic model: the classical budding yeast *S. cerevisiae*. Cells were exposed to the compounds at concentrations ranging from 2.6 to 3000 µM for 24 h. The OD600 was measured to assess cell growth. No compound showed significant bioactivity; **4q** and **4s** were the only ones showing a slight measurable effect (Appendix A). These resistances could be explained by the presence of the fungal cell wall, which could limit the entry of bis(spiropyrazolone)cyclopropanes **4** into the cell. Other mechanisms cannot be ruled out, such as structural differences in molecular targets.

Finally, a drop test assay was performed in different yeast strains (Appendix A). The sensitivity was determined in wild type yeast strain (W303 genetic background) as well as in strains harboring mutations in checkpoint pathways genes as well as those involved in signaling DNA damage lesions and/or DNA replication blocking situations. Frequently mutating strains are more sensitive to drug exposure than the wild type. This strategy hints of the mechanism of action of the analyzed drugs. The stock solutions were prepared in DMSO due to the low solubility of the compounds in ethanol. However, fresh solutions were prepared for each experiment and used immediately to avoid decomposition problems. Serial dilutions of yeast cells suspensions were spotted onto plates with compounds **4a-u** to 0.2 mg/mL (from a fresh stock in DMSO at 4 mg/mL) and incubated at 28 °C for 72 h. As a plate without drug, a plate with DMSO and a plate with genotoxic MMS 0.033% were included as controls. Control without drugs and with DMSO did not show any difference in growth indicating that DMSO is not toxic at these concentrations. However, some strains are clearly unable to grow in the presence of MMS. Our results show that neither the wild type nor the mutant strains are sensitive to any of the bis(spiropyrazolone)cyclopropanes **4** and could grow without major problems.

These observations suggest that bis(spiropyrazolone)cyclopropanes **4** do not generate DNA damage or checkpoint activation in budding yeast. It is plausible that the entry of the compounds is limited by the fungal cell wall in agreement with the high resistance observed for wild type and for mutants (Appendix A, Appendix A, see Section 3).

### 2.3. Evaluation of Cytotoxic and Leishmanicidal Activity

We next evaluated the leishmanicidal activity of the bis(spiropyrazolone)cyclopropanes **4**. The activity was evaluated in vitro against promastigotes of *Leishmania mexicana* and cytotoxicity against RAW 264.7 macrophages. Serial dilutions were immediately made in the culture medium from a fresh stock in DMSO prepared immediately before use to prevent degradation. A range of solutions from 100 to 0.001 µM were prepared for the assay. All compounds showed very good anti-leishmanicidal activity with IC_50_ values below 1 µM, except **4e** and **4f,** that have an IC_50_ value of 1.48 µM and 2.23 µM, respectively. Compounds **4b-d** and **4r-s** are the most active against leishmania with an IC_50_ ranging from 0.15 µM to 0.19 µM. These are statistically comparable to values observed for amphotericin B (0.17 µM) (see Statistical Analyses S1).

No evidence of degradation of the compounds was observed in the solutions used for treatments. The stability of 100 μM solutions of **4b** in DMSO and PBS 1X pH 7.4 was evaluated using spectroscopic measurements (Appendix A), obtaining an approximate half-life for the solution of **4b** in DMSO at room temperature for 7.85 h (471 min). On the other hand, the solution of **4** in PBS at the same conditions showed no evidence of degradation. Next, stock solutions underwent in situ isomerization to confirm that the activity is due to spyrocyclopropanes **4** and not due to their products of thermal degradation. Here, each stock solution of **4** in DMSO was heated to 100 °C for 5 min prior to the biological evaluation [41]. No effect was observed with any compounds after treating the parasite with 10 μM concentration solutions (Appendix A).

Next, we calculated the selectivity index (SI) (Table 3), and ranges from 4.9 to 45.3; **4s** has the highest SI. In the case of **4b, 4c** and **4d**, there is an SI of 21.3, 15.3 and 26.0, respectively. Compounds will be more effective and stable during treatment with a higher selectivity index.

Analysis of the substituents shows that compounds bearing electron-withdrawing groups are more active than compounds with electron-donating ones independent of their location on the ring. In compounds **4c**, **4d** and **4k**, all bearing a nitro group, the activity decreases in the order *para* > *ortho* = *meta*, but there is not enough information to make a generalization regarding the location of the substituent and its effect on the leishmanicidal activity. Regarding cytotoxicity, a pattern similar to that found in leishmanicidal activity is observed.

The results shown here indicate that bis(spiropyrazolone)cyclopropanes **4** evaluated here is potentially useful against *L. mexicana*. Compounds **4r-s** exhibit in vitro activity and a selectivity index comparable to Amphotericin B. Compounds bearing electron-withdrawing groups are more active than compounds with electron-donating ones.

### 2.4. Evaluation of Other Activities: Antioxidant and Genotoxicity

The antioxidant activity of all synthesized bis(spiropyrazolone)cyclopropanes **4** was evaluated by the *N*,*N*-diphenyl-*N*’-picrylhydrazyl (DPPH) assay (Appendix A). The radical scavenging activity of all compounds was compared with ascorbic acid used as standard. The results show that no radical scavenging activity of any of the compounds evaluated was observed, only **4k** showed activity below 300 µM.

We quantified DNA breaks by a comet assay to determine if these compounds are genotoxic. Here, *L. mexicana* cells were exposed for 24 h to MIC concentrations previously determined (see Materials and Methods Section) for each compound. Figure 1 represents the measurements obtained for tail length. All compounds generated DNA breaks expect **4s**. Compounds **4u** and **4q** had the highest genotoxic activity followed by **4k** and **4t**. Samples with DMSO and MMS (0.011%) were included as controls. Some representative pictures are shown at Appendix A. The effect of **4p** could not be determined because the cells appear to be very damaged. The data was subjected via a Kruskal–Wallis one-way ANOVA (Statistical Analyses S2). Comet assay results are consistent with leishmanicidal activities showing the biggest genotoxic effects after exposure to **4k**, **4q**, **4t** and **4u** compounds. These observations support a mechanism of action through the generation of DNA damage.

Important limitations to the actual effectiveness of any compound as a therapeutic agent are the safety of a compound in terms of cytotoxicity and genotoxicity towards mammal cells. To address genotoxicity we calculated in silico the theoretical prediction of ADME (adsorption, distribution, metabolism and excretion) properties of compounds **4b-u** (Table 4).

To evaluate the drug-likeness of a compound, Lipiniski’s rule of five (ROF) is usually used, as we did previously [38]. ROF states that good absorption or permeation is more likely when: (a) there are no more than five hydrogen bond donors (HBD); (b) no more than 10 hydrogen bond acceptors (HBA); (c) formula weight less than 500; and (d) n-octanol/water partition coefficient (log P) less than 5. Two or more violations of ROF suggest the probability of problems in bioavailability [43]. One of the big problems with the development of new drugs is that they usually fail before reaching clinics due to poor pharmacokinetics, and these physicochemical drug descriptors provide a useful tool for evaluating drug activity. We calculated these descriptors for the compounds **4b-u** using the Osiris DataWarrior software (Table 4) [44]. ADME predictions suggest good bioavailability of compounds **4b-k**, with small limitations of **4p-u**. The compounds **4r** and **4s** have the worst predictions, with two violations of ROF. In addition, no compound shows a predicted behavior of mutagenicity, tumorigenicity, reproductive or irritant effect, with the one exception of **4q**, which presents a low irritant effect. Overall, the ADME scores of bis(spiropyrazolone)cyclopropanes **4-u** are quite satisfactory.

We wanted to evaluate the genotoxic effect of compounds in mammal cells. Since human cancer cell lines are resistant to bis(spiropyrazolone)cyclopropanes **4**, we reasoned that no genotoxic effect should be observed after exposure to the compounds. To confirm this, we conducted an in vitro comet assay in a non-cancerous mammal cell line, CHO-K1. The viability measured by FDA/Ethidium Bromide, for all the evaluated compounds was greater than 93%. Statistical analyses can be consulted in the Appendix A (Statistical Analyses S3). The compounds **4b**, **4q**, **4t**, and **4u** did not show differences with respect to the control of DMSO. The most significant were **4k** and **4d** (Figure 2).

## 3. Discussion

In this study, we synthesized a series of 12 bis(spiropyrazolone)cyclopropanes **4** and tested their cytotoxic properties against selected yeast, human cancer cell lines and leishmania parasites. Additionally, we evaluated their antioxidant and genotoxic properties.

The results point to a specific activity against leishmania parasites with SI values comparable to Amphotericin B, a first line drug used in leishmaniasis treatment. The results also indicate that human cancer cell lines and budding yeast are resistant to these compounds, suggesting different modes of action or different entry barriers.

Leishmaniasis is a neglected tropical disease that affects humans and some animals and is transmitted by the bite of sandflies infected with a protozoan parasite of the genus *Leishmania*. There are three main clinical forms of this disease depending on the parasite, immune-inflammatory responses, and the immune status of the host. Chronic infections become an acute life-threatening condition in immunocompromised individuals [45]. The disease spectrum of leishmaniasis goes from (i) a subclinical form (not apparent) or (ii) a localized (skin lesion), to (iii) a disseminated mucocutaneous and visceral form (the most important and deadly disease) [46,47].

Leishmaniasis affects the most economically vulnerable populations, and thus pharmaceutical companies have been reluctant to make important investments for the development of new treatments. This fact highlights the importance of making efforts investigating new alternatives for its treatment. An interesting approach is the use of compounds that interfere with DNA replication or metabolism that have also been frequently used as anti-tumor agents taking advantage of the characteristic of tumor cells and parasites of active replication. A comprehensive review reveals multiple differences regarding DNA repair and replication machinery between parasites of the *Leishmania* genus and mammalian cells [48], thus suggesting that these proteins may be potential targets for fighting leishmaniasis, which has been under-explored until now.

Leishmaniasis treatment is currently based on the use of a few drugs, mostly parenteral. These drugs are expensive with limited effectiveness. They are also long-lasting with severe adverse effects generating a high dropout rate. The medications normally used to treat visceral leishmaniasis are systemic agents such as pentavalent antimony compounds (Pentostam (Brentford, UK), Glucantime^®^, (Paris, France), as well as Amphotericin B, pentamidine, paromomycin, and miltefosine (the only oral drug). Patients in high-income countries are treated with liposomal Amphotericin B (AmBisome^®^, San Dimas, CA, USA). In the case of cutaneous leishmaniasis, local and/or systemic applications of the same previous drugs are used, as well as pentamidine and ketoconazole [47]. The most serious drawbacks with which health professionals must deal in the treatment of leishmaniasis with these drugs are their high toxicity, presence of severe adverse effects, generation of resistance and inconsistency in their effectiveness against different species of *Leishmania* parasites [49].

Our results indicate that selected cancer cell lines and yeast are not sensitive to the bis(spiropyrazolone)cyclopropanes **4** (Appendix A). Yeasts are especially resistant to these compounds, even already sick double mutants, such as *rad9 tof1* (Appendix A). These do not display major growth problems in the presence any of the bis(spiropyrazolone)cyclopropanes **4**. This remarkable resistance could be due to two possibilities: (i) absence of a biological target, such as an enzyme; (ii) the cell wall and/or membrane acting as natural barrier [50,51]. The cell wall has a sieve-like structure, and usually does not limit the transport of biomolecules. However, the cell wall can limit the entry of some compounds such as bleomycin or tetraphenylphosphonium in some cases [50,51].

The plasma membrane is the other structure that could be involved in resistance. It limits the entry of small molecules by simple diffusion and controls the entry of bigger ones by transporters and channels. Drug resistance can be developed by several microorganisms associated with ATP-binding cassettes (ABC) transporters, efflux pumps or lipidome changes [52]. Nevertheless, this resistance is developed after exposure periods. It is possible to increase the cell wall and plasma membrane permeability with organic solvents, detergents, or exposure to an electric field [50,53]. In the case of the resistance observed in *S. cerevisiae* to the bis(spiropyrazolone)cyclopropanes **4** it is more likely to consider a preexistent natural barrier because there is no previous exposure to these compounds.

On the other hand, the assayed human cell lines also showed resistance to the bis(spiropyrazolone)cyclopropanes **4**, but in a lower order of magnitude possibly explained by the absence of cell wall (Table 2). In any case, the IC_50_ values are still too elevated to be exploitable in terms of anticancer therapeutics. Strikingly, the compounds showed a high activity against *L. mexicana* (Table 3). The calculated SI gives relatively good ratios. Amphotericin B was used as a control because it is a first line drug used in leishmaniasis treatment. The less active compounds are **4e** and **4f** with an IC_50_ statistically different from Amphotericin B (groups a and b, respectively). Compounds **4r** and **4s** exhibit very good leishmanicidal activity with an IC_50_ of 0.19 µM for both. This is close to the IC_50_ (0.17 µM) of Amphotericin B, all of them belonging to group d. The SI ratios of **4r** and **4s** are 34.2 and 45.3, respectively. In contrast to the bis(spiropyrazolone)cyclopropanes **4**, Amphotericin B is also effective against yeasts and fungi [54]. This observation, however, cannot distinguish if this difference is due to a different mechanism of action, or to a natural resistance specific for bis(spiropyrazolone)cyclopropanes **4**. More analysis remains to be done to determine if compounds **4r** and **4s** can be real therapeutic agents for leishmaniasis, but preliminary results encourage us to move forward next to in vivo studies.

Compounds **4b**, **4c** and **4d** are also classified into group d and are the most active with an IC_50_ of 0.16 µM for **4b** and 0.15 µM both for **4c** and **4d**. Substitution on the aromatic ring decreases the activity versus **4b** independent of the nature of the substituent; however, no effect was observed in the case of **4c** and **4d**—these both have a nitro group at *ortho* and *meta* positions, respectively. Analysis of the substituents shows that compounds bearing electron-withdrawing groups are more active than compounds with electron-donating ones independent of their location on the ring. Nevertheless, there is not enough information to make a generalization regarding the location of the substituent and its effect on the leishmanicidal activity.

We performed comet assay analyses in leishmania cells exposed to the compounds to better understand the underlying mechanism of action of bis(spiropyrazolone)cyclopropanes **4**. This approach gives a measure of the presence of DNA damage after drug exposure. We performed the analyses in alkaline conditions, to detect DNA double-strand breaks, single-strand breaks, and alkali-labile sites [55,56]. The results showed the presence of DNA damage to a greater or lesser extent after exposure to the compounds (Figure 1). The strongest effects were induced by **4k**, **4q** and **4u**, while the weakest was generated by **4s**. While genotoxic effect can somehow explain the mechanism of action in leishmania cells, it is undesirable if it occurs in normal mammal cell lines. To evaluate this, comet assay was performed in the CHO-K1 cell line (Figure 2). This cell line presented only 6% inhibition (Appendix A), and comet assay was performed in cells with a viability of at least 93%. Analyses indicate less effect in the tail length of CHO K-1 cells population exposed to bis(spiropyrazolone)cyclopropanes **4** (Figure 2) with respect to *L. mexicana* cells. In addition, theoretical prediction of adsorption, distribution, metabolism, and excretion (ADME) properties of bis(spiropyrazolone)cyclopropanes **4** returns good scores with only minor rules violation. Interestingly, no undesired effects (mutagenicity, tumorigenicity, reproductive or irritant effects) are described for these compounds, except **4q**, which has a low predicted irritant effect.

## 4. Materials and Methods

### 4.1. Chemistry

#### 4.1.1. General

All solvents and reagents were from Aldrich (St. Louis, MO, USA) and used without further purification. All melting points are uncorrected and were determined on a Büchi Melting Point M-560 apparatus (Büchi Labortechnik AG, Flawil, Switzerland). FTIR spectra were recorded by a Perkin Elmer FTIR Spectrum One (Waltham, MA, USA) by using ATR system (4000–650 cm^−1^). The ^1^H and ^13^C NMR spectra were recorded at 298 K on a Varian 400/54 Premium Shielded NMR Magnet System (Yarnton, Oxford, UK USA) using CDCl_3_ and DMSO-*d_6_* as solvents. The ^19^F-NMR spectra were acquired on an Oxford Instruments Pulsar benchtop NMR 60 MHz Spectrometer (Tubney Woods, Abingdon, Oxford, UK). Chemical shifts are expressed in ppm with TMS as an internal reference (TMS, δ = 0 ppm) for protons and trifluoroacetic acid (TFA, δ = −75.39 ppm) for fluorine Accurate mass data was obtained using a Waters (Waltham, MA, USA) model LCT Premiere time-of-flight (TOF) mass spectrometer. Reactions were monitored by TLC on silica gel using ethyl acetate/hexane mixtures as a solvent and compounds visualized by UV lamp. The reported yields are for the purified material and are not optimized.

#### 4.1.2. General Procedure for the Synthesis of 4,4′-(Arylmethylene)bis(3-methyl-1-phenyl-1*H*-pyrazol-5-ols) **2a-u**

All 4,4′-(arylmethylene)bis(3-methyl-1-phenyl-1*H*-pyrazol-5-ols) **2**, except **2h** and **2m**, were reported previously [12,57,58,59] and synthesized according to the reported procedures by our research group [12]. To a solution of 0.4 mmol aldehyde **3a-u** and 0.8 mmol pyrazol **1** in 4 mL of 70% EtOH, 40.2 μL of 1 M NaOAc were added and the mixture was stirred at room temperature until the reaction was complete. Water was added to obtain 50% EtOH and the mixture was filtered, washed with 50% EtOH and dried to obtain pure product.

*4,4′-[(5-Hydroxy-2-nitrophenyl)methylene]bis(3-methyl-1-phenyl-1H-pyrazol-5-ol)* (**2h**): yield 93% as a slightly yellow solid; mp 205–207 °C (d); ^1^H-NMR (500 MHz, DMSO-*d_6_*) δ: 13.23 (br. s., 1 H), 10.65 (s, 1 H), 7.72 (d, *J* = 8.8 Hz, 1 H), 7.68 (d, *J* = 7.7 Hz, 4 H), 7.44 (t, *J* = 7.7 Hz, 4 H), 7.24 (t, *J* = 7.1 Hz, 2 H), 7.06–7.01 (m, 1 H), 6.75 (dd, *J* = 2.7, 8.8 Hz, 1 H), 5.61 (s, 1 H), 2.24 (s, 6 H); ^13^C-NMR (126 MHz, DMSO-*d_6_*) δ: 160.9, 146.1, 141.0, 138.2, 129.0, 127.6, 125.7, 125.7, 125.7, 120.5, 117.0, 113.6, 29.8, 11,6; FTIR (cm^−1^): 3615, 2924, 1595, 1498, 1349, 1258, 1070, 751, 685; HRMS (TOF ES+) *m*/*z* calcd for C_27_H_24_N_5_O_5_ (M + H)^+^: 498.1772; found: 498.1777.

*4,4′-[(4-Hydroxy-3,4-dimethoxyphenyl)methylene]bis(3-methyl-1-phenyl-1H-pyrazol-5-ol)* (**2m**): yield 97% as a cream solid; mp 208–210 °C; ^1^H-NMR (500 MHz, DMSO-*d_6_*) δ: 14.10 (s, 1 H), 12.41 (br. s., 1 H), 8.19 (br. s., 1 H), 7.70 (d, *J =* 8.23 Hz, 4 H), 7.44 (t, *J =* 7.68 Hz, 4 H), 7.24 (t, *J =* 7.41 Hz, 2 H), 6.61 (s, 2 H), 4.84 (s, 1 H), 3.67 (s, 6 H), 2.32 (br. s., 6 H); ^13^C-NMR (126 MHz, DMSO-*d_6_*) δ: 147.7, 146.1, 134.2, 132.8, 128.9, 125.6, 120.7, 105.3, 56.1, 33.4, 11.8; FTIR (cm^−1^): 3225, 1607, 1580, 1505, 1454, 1223, 1107, 762, 698; HRMS (TOF ES+) *m*/*z* calcd for C_29_H_29_N_4_O_5_ (M + H)^+^: 513.2132; found: 513.2138.

#### 4.1.3. Synthesis of (5R*,6R*)-11-Aryl-4,10-dimethyl-2,8-diphenyl-2,3,8,9-tetraazadispiro[4.0.4.1]undeca-3,9-diene-1,7-diones **4**

The synthesis of the spirocyclopropanes **4** was adapted from a procedure reported by Elinson and coworkers [34]. To a 1 millimolar solution of **2** in MeOH was added 0.6 mmol of NaBr and the mixture was electrolyzed at room temperature in an undivided cell equipped with magnetic stirrer. The electrochemical device used for the reaction was ensembled as previously reported [60], using a 6 V, 800 mA D.C. power supply, a 2.0 mm mechanical pencil lead refills as graphite anode, and an iron wire with the same diameter as cathode (electrodes surface 1.2 cm^2^). Current was passed through the reaction mixture until starting material was consumed (sometimes it is necessary to clean both electrodes to withdraw the spots of solid deposited). The progress of the reaction was monitored by TLC. When the electrolysis was finished, the mixture was gently concentrated under vacuum to one fifth of initial volume and stored at 0 °C overnight. The precipitates formed were collected by filtration, rinsed with the minimum amount of ice-cold MeOH and then dried in vacuum to produce the pure compounds.

(5*R**,6*R**)-*4,10-Dimethyl-2,8,11-triphenyl-2,3,8,9-tetraazadispiro[4.0.4.1]undeca-3,9-diene-1,7-dione* (**4b**): yield 81% as a white solid; mp 129–130 °C [Lit 166–168 °C] [34]; ^1^H-NMR (400 MHz, CDCl_3_) δ: 2.09 (s, 3 H), 2.53 (s, 3 H), 4.45 (s, 1 H), 7.18–7.26 (m, 4 H), 7.36–7.47 (m, 7 H), 7.89 (dd, *J* = 8.6, 1.2 Hz, 2 H), 7.93 (dd, *J* = 8.6, 1.2 Hz, 2 H); ^13^C-NMR (100 MHz, CDCl_3_) δ: 18.5, 20.4, 43.2, 50.4, 51.4, 119.1, 125.6, 125.7, 128.1, 128.7, 128.9, 129.0, 129.1, 130.0, 137.9, 138.0, 155.4, 156.1, 165.7, 167.8; FTIR (cm^−1^): 1707, 1595, 1499, 744, 688; ESI-MS, m/z, (rel. int.): 435.10 (55.22) [M + H]^+^,262.94 (16.14), 174.84 (100.0).

(5*R**,6*R**)-*11-(2-Nitrophenyl)-4,10-dimethyl-2,8-diphenyl-2,3,8,9-tetraazadispiro[4.0.4.1]undeca-3,9-diene-1,7-dione* (**4c**): yield 70% as a white solid; mp 165.1–165.5 °C; ^1^H-NMR (400 MHz, CDCl_3_) δ: 2.05 (s, 3H), 2.58 (s, 3H), 4.66 (bs, 1H), 7.20 (t,t, *J* = 7.5, 1.2 Hz 1H), 7.25 (t,t, *J* = 7.5, 1.2 Hz 1H), 7.35 (ddd, *J =* 8.9, 1.4, 1.0 Hz, 1 H), 7.38 (dd, *J =* 8.6, 7.4 Hz, 2 H), 7.45 (dd, *J =* 8.6, 7.4 Hz, 2 H), 7.60 (dddd, *J =* 8.9, 6.9, 1.4, 0.8 Hz, 1 H), 7.68 (ddd, *J =* 8.2, 6.9, 1.4 Hz, 1 H), 7.75 (dd, *J =* 8.6, 1.2 Hz, 2 H), 7.94 (dd, *J =* 8.6, 1.2 Hz, 2 H), 8.17 (dd, *J =* 8.2, 1.4 Hz, 1 H); ^13^C-NMR (100 MHz, CDCl_3_) δ: 18.3, 20.3, 42.1, 50.5, 51.2, 119.2, 119.6, 125.1, 125.6, 125.9, 129.0, 129.2, 130.1, 133.1, 133.5, 137.7, 137.8, 149.1, 154.1, 156.4, 165.4, 165.7, 167.4; FTIR (cm^−1^): 1702, 1595, 1521, 1498, 1347, 758, 689; ESI–MS, *m*/*z*, (rel. int.): 502.18 (100) [M + Na]^+^.

(5*R**,6*R**)-*11-(3-Nitrophenyl)-4,10-dimethyl-2,8-diphenyl-2,3,8,9-tetraazadispiro[4.0.4.1]undeca-3,9-diene-1,7-dione* (**4d**): yield 91% as a white solid; mp 183.9–185.0 °C; ^1^H-NMR (400 MHz, CDCl_3_) δ: 2.09 (s, 3 H), 2.53 (s, 3 H), 4.45 (s, 1 H), 7.22 (t, *J =* 7.5 Hz, 1H), 7.26 (t, *J =* 7.5 Hz, 1H), 7.41 (dd, *J =* 8.6, 7.5 Hz, 2 H), 7.46 (dd, *J =* 8.6, 7.5 Hz, 2 H), 7.53–7.62 (m, 2 H), 7.84 (dd, *J =* 8.4, 1.2 Hz, 2 H), 7.93 (dd, *J =* 8.4, 1.2 Hz, 2 H), 8.09–8.12 (m, 1 H), 8.26 (dt, *J =* 7.5, 2.0 Hz, 1 H); ^13^C-NMR (100 MHz, CDCl_3_) δ: 18.5, 20.4, 41.7, 49.8, 50.8, 119.1, 119.2, 123.9, 125.2, 125.9, 126.0, 129.1, 129.2, 129.8, 130.6, 136.1, 137.7, 148.3, 154.1, 155.6, 165.3, 167.2; FTIR (cm^−1^): 1702, 1595, 1532, 1488, 1365, 845, 759, 691; ESI–MS, *m*/*z*, (rel. int.): 480.23 (39.01) [M + H]^+^, 172.82 (100).

(5*R**,6*R**)-*11-(4-Methoxyphenyl)-4,10-dimethyl-2,8-diphenyl-2,3,8,9-tetraazadispiro[4.0.4.1]undeca-3,9-diene-1,7-dione* (**4e**): yield 87% as a white solid; mp 163.5–165.7 °C; ^1^H-NMR (400 MHz, CDCl_3_) δ: 2.10 (s, 3 H), 2.53 (s, 3 H), 3.82 (s, 3 H), 4.39 (bs, 1 H), 6.90 (d, *J* = 8.8 Hz, 2 H), 7.12 (dd, *J* = 8.8, 0.8 Hz, 2 H), 7.20 (tt, *J* = 7.4, 1.2 Hz 1 H), 7.23 (tt, *J* = 7.4, 1.0 Hz 1 H), 7.40 (dd, *J* = 8.5, 7.5 Hz 2 H), 7.44 (dd, *J* = 8.5, 7.5 Hz 2 H), 7.89 (dd, *J* = 8.8, 1.0 Hz, 2 H), 7.93 (dd, *J* = 8.8, 1.0 Hz, 2 H); ^13^C-NMR (100 MHz, CDCl_3_) δ: 18.3, 20.2, 42.7, 50.4, 51.5, 55.2, 114.0, 118.9, 119.7, 125.4, 125.5, 128.8, 128.9, 131.0, 137.8, 137.8, 155.3, 156.0, 159.7, 165.6, 167.7; FTIR (cm^−1^): 1706, 1593, 1589, 1251, 1035, 757, 689; ESI–MS, *m*/*z*, (rel. int.): 465.15 (6.66) [M + H]^+^, 292.97 (59.57), 174.85 (100.0).

(5*R**,6*R**)-*11-(2-Methoxyphenyl)-4,10-dimethyl-2,8-diphenyl-2,3,8,9-tetraazadispiro[4.0.4.1]undeca-3,9-diene-1,7-dione* (**4f)**: yield 77% as a white solid; mp 147.8–149.0 °C [Lit 145–146 °C] [34]; ^1^H-NMR (400 MHz, CDCl_3_) δ: 2.11 (s, 3 H), 2.53 (s, 3 H), 3.62 (s, 3 H), 4.11 (bs, 1 H), 6.89 (d, *J* = 8.4 Hz, 1 H), 6.98 (td, *J* = 7.4, 0.8 Hz 1 H), 7.13 (dt, *J* = 7.4, 1.3 Hz, 1 H), 7.18 (tt, *J* = 7.4, 1.2 Hz 1 H), 7.23 (tt, *J* = 7.4, 1.2 Hz 1 H), 7.34–7.41 (m, 3 H), 7.44 (dd, *J* = 8.6, 7.4 Hz 2 H), 7.87 (dd, *J* = 8.6, 1.2 Hz, 2 H), 7.94 (dd, *J* = 8.6, 1.2 Hz, 2 H); ^13^C-NMR (100 MHz, CDCl_3_) δ: 18.5, 20.3, 39.5, 50.7, 51.1, 55.6, 110.8, 116.6, 119.0, 119.2, 120.5, 125.3, 125.6, 129.0, 129.1, 130.3, 131.2, 138.0, 138.2, 155.9, 156.4, 157.8, 166.1, 168.1. FTIR (cm^−1^): 1708, 1593, 1489, 1296, 777, 753; LC–MS, *m*/*z*, (rel. int.): 466.48 (14.86) [M + H]^+^, 292.96 (50.29), 174.84 (100.0).

(5*R**,6*R**)-*11-(4-Nitrophenyl)-4,10-dimethyl-2,8-diphenyl-2,3,8,9-tetraazadispiro[4.0.4.1]undeca-3,9-diene-1,7-dione* (**4k**): yield 54% as a white solid; mp 145.3–152.9 (d) °C; ^1^H-NMR (400 MHz, CDCl_3_) δ: 2.09 (s, 3 H), 2.53 (s, 3 H), 4.44 (bs, 1 H), 7.22 (tt, *J* = 7.4, 1.2 Hz, 1 H), 7.26 (tt, *J* = 7.4, 1.2 Hz, 1 H), 7.40 (dd, *J* = 9.0, 1.1 Hz, 2 H), 7.41 (dd, *J* = 8.6, 7.4 Hz, 2 H), 7.45 (dd, *J* = 8.6, 7.4 Hz, 2 H),7.85 (dd, *J* = 8.6, 1.2 Hz, 2 H), 7.92 (dd, *J* = 8.6, 1.2 Hz, 2 H), 8.25 (d, *J* = 9.0 Hz, 2 H); ^13^C-NMR (100 MHz, CDCl_3_) δ: 18.5, 20.5, 41.9, 49.9, 50.8, 119.1, 119.1, 123.9, 125.9, 126.0, 129.1, 129.2, 131.2, 135.8, 137.68, 137.72, 148.1, 154.2, 155.6, 165.3, 167.2; FTIR (cm^−1^): 1708, 1594, 1489, 1347, 1296, 756, 688.5; LC–MS, *m*/*z*, (rel. int.): 502.18 (100) [M + Na]^+^.

(5*R**,6*R**)-*11-(4-Fluorophenyl)-4,10-dimethyl-2,8-diphenyl-2,3,8,9-tetraazadispiro[4.0.4.1]undeca-3,9-diene-1,7-dione* (**4p**): yield 47% as a white solid; mp 128.9–129.9 °C; ^1^H-NMR (400 MHz, CDCl_3_) δ: 2.09 (s, 3 H), 2.52 (s, 3 H), 4.37–4.40 (m, 1 H), 7.08 (dd, *J* = 8.6, 8.5 Hz, 2 H), 7.19 (ddd, J = 9.0, 5.1, 0.8 Hz, 2 H), 7.21 (tt, J = 7.4, 1.2 Hz, 1 H), 7.24 (tt, *J* = 7.4, 1.2 Hz, 1 H), 7.41 (dd, *J* = 7.5, 8.6 Hz, 2 H), 7.44 (dd, *J* = 7.5, 8.6 Hz, 2 H), 7.87 (dd, *J* = 8.8, 1.0 Hz, 2 H), 7.92 (dd, *J* = 8.6, 1.2 Hz, 2 H); ^13^C-NMR (100 MHz, CDCl_3_) δ: 18.5, 20.4, 42.3, 50.3, 51.4, 115.9 (d, *J* = 21.8 Hz, 2 C), 119.1, 123.9 (d, *J* = 3.1 Hz, 1 C), 125.7, 125.8, 129.1, 129.1, 131.8 (d, *J* = 8.6 Hz, 2 C), 137.8, 137.9, 155.0, 156.0, 162.9 (d, *J* = 249.1 Hz, 1 C), 165.6, 167.7; ^19^F-NMR (56.17 MHz; *CDCl_3_*) δ: −112.46 (bs); FTIR (cm^−1^): 1712, 1595, 1499, 1233, 756, 689; LC–MS, *m*/*z*, (rel. int.): 475.35 (100) [M + Na]^+^, 453.4 (92.04) [M + H]^+^.

(5*R**,6*R**)-*Methyl-4[4,10-dimethyl-2,8-diphenyl-2,3,8,9-tetraazadispiro[4.0.4.1]undeca-3,9-diene-1,7-dione-11-yl]benzoate* (**4q**): yield 85% as a white solid; mp 162.5–163.9 °C; ^1^H-NMR (400 MHz, CDCl_3_) δ: 2.08 (s, 3 H), 2.53 (s, 3 H), 3.93 (s, 3 H), 4.44 (bs, 1 H), 7.21 (tt, *J* = 7.4, 1.2 Hz, 1 H), 7.24 (tt, *J* = 7.4, 1.2 Hz, 1 H), 7.30 (dd, *J* = 8.3, 0.8 Hz, 2 H), 7.40 (dd, *J* = 8.6, 7.4 Hz, 2 H), 7.44 (dd, *J* = 8.6, 7.4 Hz, 2 H), 7.87 (dd, *J* = 8.6, 1.2 Hz, 2 H), 7.93 (dd, *J* = 8.6, 1.2 Hz, 2 H), 8.06 (d, *J* = 8.3 Hz, 2 H); ^13^C-NMR (100 MHz, CDCl_3_) δ: 18.5, 20.4, 42.6, 50.1, 51.1, 52.4, 119.1, 119.1, 125.7, 125.8, 129.1, 129.2, 130.0, 130.2, 130.7, 133.4, 137.8, 137.9, 154.9, 155.8, 165.4, 166.5, 167.5; FTIR (cm^−1^): 1709, 1595, 1494, 1279, 759, 694; LC–MS, *m*/*z*, (rel. int.): 515.28 (6.77) [M + Na]^+^, 493.32 (100) [M + H]^+^.

(5*R**,6*R**)-*11-(4-Trifluoromethylphenyl)-4,10-dimethyl-2,8-diphenyl-2,3,8,9-tetraazadispiro[4.0.4.1]undeca-3,9-diene-1,7-dione* (**4r)**: yield 87% as a white solid; mp 154.6–156.5 °C; ^1^H-NMR (400 MHz, *CDCl_3_*) δ: 2.08 (s, 3 H), 2.53 (s, 3 H), 4.43 (d, *J* = 0.9 Hz, 1 H), 7.22 (tt, *J* = 7.4, 1.2 Hz, 1 H), 7.25 (tt, *J* = 7.4, 1.2 Hz, 1 H), 7.34 (d, *J* = 8.3 Hz, 2 H), 7.41 (dd, *J* = 8.7, 7.4 Hz, 2 H), 7.45 (dd, *J* = 8.7, 7.4 Hz, 2 H), 7.65 (d, *J* = 8.3 Hz, 2 H), 7.86 (dd, *J* = 8.7, 1.2 Hz, 2 H), 7.93 (dd, *J* = 8.8, 1.2 Hz, 2 H); ^13^C-NMR (100 MHz, *CDCl_3_*) δ: 18.5, 20.4, 42.2, 50.0, 51.0, 119.1, 123.9 (q, *J* = 272.5 Hz, 1 C), 125.7 (q, *J* = 3.9 Hz, 2 C), 125.8, 125.9, 129.1, 129.2, 130.5, 131.1 (q, *J* = 32.7 Hz, 1 C), 132.4 (q, *J* = 1.2 Hz, 2 C), 137.78, 137.82, 154.7, 155.8, 165.4, 167.5; ^19^F-NMR (56.17 MHz; *CDCl_3_*) δ: −62.71 (s); FTIR (cm^−1^): 1704, 1595, 1498, 1321, 1110, 751, 687; LC–MS, m/z, (rel. int.): 525.33 (62.46) [M + Na]^+^, 503.36 (100) [M + H]^+^, 413.47 (16.18).

(5*R**,6*R**)-*11-(4-Trifluoromethoxyphenyl)-4,10-dimethyl-2,8-diphenyl-2,3,8,9-tetraazadispiro[4.0.4.1]undeca-3,9-diene-1,7-dione* (**4s**): yield 70% as a white solid; mp 150.5–151.5 °C; ^1^H-NMR (400 MHz, CDCl_3_) δ: 2.08 (s, 3 H), 2.52 (s, 3 H), 4.39 (bs, 1 H), 7.19–7.27 (m, 6 H), 7.41 (dd, J = 8.6, 7.4 Hz, 2 H), 7.45 (dd, J = 8.6, 7.4 Hz, 2 H), 7.87 (dd, J = 8.8, 1.2 Hz, 2 H), 7.92 (dd, J = 8.6, 1.2 Hz, 2 H); ^13^C-NMR (100 MHz, *CDCl_3_*) δ: 18.5, 20.4, 42.1, 50.1, 51.3, 119.1, 119.2, 121.1, 120.5 (q, *J* = 257.7 Hz, 1 C), 125.75, 125.82, 126.8, 129.1, 129.2, 131.6, 137.8, 137.9, 149.5 (d, *J* = 1.6 Hz, 1 C), 154.8, 155.9, 165.5, 167.6; ^19^F-NMR (56.17 MHz; CDCl_3_) δ: −57.91 (s); FTIR (cm^−1^): 1705, 1594, 1498, 1254, 1222, 1166, 750, 690; LC–MS, *m*/*z*, (rel. int.): 541.39 (100) [M + Na]^+^, 519. 46 (90.66) [M + H]^+^.

(5*R**,6*R**)-*11-(3-Fluorophenyl)-4,10-dimethyl-2,8-diphenyl-2,3,8,9-tetraazadispiro[4.0.4.1]undeca-3,9-diene-1,7-dione* (**4t**): yield 84% as a white solid; mp 152.1–153.0 °C; ^1^H-NMR (400 MHz, CDCl_3_) δ: 2.12 (s, 3 H), 2.52 (s, 3 H), 4.40 (s, 1 H), 6.94 (dddd, *J* = 9.2, 2.5, 1.2, 1.2 Hz, 1 H), 7.00 (dddd, *J* = 7.9, 1.9, 0.8, 0.8 Hz 1 H), 7.09 (ddddd, *J* = 8.7, 8.2, 2.6, 0.8, 0.8 Hz 1 H), 7.21 (tt, *J* = 7.4, 1.2 Hz, 1 H), 7.24 (tt, *J* = 7.4, 1.2 Hz, 1 H), 7.36 (ddd, *J* = 8.3, 7.8, 5.9 Hz, 1 H), 7.41 (dd, J = 8.6, 7.4 Hz, 2 H), 7.44 (dd, J = 8.6, 7.4 Hz, 2 H), 7.87 (dd, J = 8.8, 1.2 Hz, 2 H), 7.92 (dd, J = 8.8, 1.2 Hz, 2 H); ^13^C-NMR (100 MHz, CDCl_3_) δ: 18.3, 20.2, 42.2 (d, J = 2.3 Hz, 1 C), 50.0, 51.0, 115.8 (d, *J* = 21.1 Hz, 1 C), 117.0 (d, *J* = 22.6 Hz, 1 C), 118.97, 118.98, 125.5, 125.6, 125.6 (d, *J* = 3.1 Hz, 1 C), 128.9, 129.0, 130.2 (d, *J* = 8.6 Hz, 1 C), 130.4 (d, *J* = 8.6 Hz, 1 C), 137.66, 137.73, 154.7, 155.7, 162.5 (d, *J* = 248.3 Hz, 1 C), 165.3, 167.4; ^19^F-NMR (56.17 MHz; *CDCl_3_*) δ: −111.90 (bs); FTIR (cm^−1^): 1714, 1594, 1499, 1134, 751, 691; LC–MS, *m*/*z*, (rel. int.): 475.21 (100) [M + Na]^+^, 453.24 (42) [M + H]^+^.

(5*R**,6*R**)-*11-(4-Thiomethylphenyl)-4,10-dimethyl-2,8-diphenyl-2,3,8,9-tetraazadispiro[4.0.4.1]undeca-3,9-diene-1,7-dione* (**4u**): yield 64% as a white solid; mp 154.4–157.3 °C; ^1^H-NMR (400 MHz, CDCl_3_) δ: 2.1 (s, 3 H), 2.5 (s, 3 H), 2.5 (s, 3 H), 4.4 (bs, 1 H), 7.11 (dd, *J* = 8.6, 0.8 Hz, 2 H), 7.21 (tt, *J* = 7.4, 1.2 Hz, 1 H), 7.23 (d, *J* = 8.3 Hz, 2 H), 7.23 (tt, *J* = 7.4, 1.2 Hz, 1 H), 7.40 (dd, *J* = 8.6, 7.4 Hz, 2 H), 7.44 (dd, *J* = 8.6, 7.4 Hz, 2 H), 7.88 (dd, *J* = 8.7, 1.2 Hz, 2 H), 7.92 (dd, *J* = 8.6, 1.2 Hz, 2 H); ^13^C-NMR (100 MHz, CDCl_3_) δ: 15.4, 18.3, 20.3, 42.6, 50.2, 51.3, 119.0, 124.3, 125.4, 125.6, 126.1, 128.9, 129.0, 130.2, 137.7, 137.8, 139.6, 155.1, 155.9, 165.5, 167.6; FTIR (cm^−1^): 1715, 1596, 1498, 1365, 1130, 754, 689; LC–MS, *m*/*z*, (rel. int.): 503.22 (100) [M + Na]^+^, 481.23 (7.62).

*4-(4-(Dimethylamino)benzylidene)-3-methyl-1-phenyl-1H-pyrazol-5(4H)-one* (**5**): yield 60% as a red needles; mp 190.7–193.2 °C [Ref 192–193 °C] [61]; ^1^H-NMR (400 MHz, CDCl_3_) δ: 2.33 (s, 3 H), 3.13 (s, 6 H), 6.73 (d, *J* = 9.0 Hz, 2 H), 7.15 (t, *J* = 7.3 Hz, 1 H), 7.26 (br s, 1H), 7.40 (t, *J* = 7.8 Hz, 2 H), 8.01 (d, *J* = 7.8 Hz, 2 H), 8.58 (m, *J* = 9.0 Hz, 2 H).

*4,10-dimethyl-2,8-diphenyl-2,3,8,9-tetraazadispiro[4.0.4^6^.1^5^]undeca-3,9-diene-1,7-dione* (**6**): yield 4% as a white solid; mp: 165,8–167,9 °C; ^1^H-NMR (400 MHz, CDCl_3_) δ: 2.42 (s, 6 H), 2.77 (s, 2 H), 7.21 (tt, *J* = 7.4, 1.2 Hz, 2 H), 7.41 (dd, *J* = 8.6, 7.4 Hz, 4 H), 7.86 (dd, *J* = 8.6, 1.2 Hz, 4 H); ^13^C-NMR (100 MHz, CDCl_3_) δ: 18.3, 24.8, 47.4, 119.2, 125.7, 129.1, 137.9, 156.3, 167.7; FTIR (cm^−1^): 1702, 1593, 1488, 1361, 1298, 755.9, 692.2; LC–MS, *m*/*z*, (rel. int.): 381.16 (100) [M + Na]^+^, 359.19 (8.9) [M + H]^+^.

### 4.2. Biological Evaluation

#### 4.2.1. Evaluation of the Fungicidal Activity on *S. cerevisiae*

The antimicrobial activities of the synthesized compounds were tested against yeast *S. cerevisiae* (background W303), as described by Teran et al. [38]. Concentrations ranging from 2468.7 to 2.4 µM were assayed.

#### 4.2.2. Drop Test

The stock solutions of the compounds were prepared by dissolving **4c**, **4k** and **4p-u** in dimethyl sulfoxide (DMSO) to a concentration of 4 mg/mL, and compounds **4b**, **4e**, **4d** and **4f** were dissolved in EtOH. The solutions were then diluted on YPD plates to a concentration of 0.1 mg/mL. DMSO 2.5% and MMS (methyl methane sulfonate) 0.025% were used as control. Drop tests were performed as described in Teran et al. [38].

#### 4.2.3. Cell Culture Procedures

Five tumor human cancer cell lines—cerebral astrocytoma (D-384), grade IV prostatic adenocarcinoma (PC-3), mammary adenocarcinoma (MCF-7), human colon carcinoma (RKO) and lung carcinoma (A-549)—along with one immortalized hamster cell line (CHO K-1) were used. The cells PC-3, MCF-7 and RKO were cultured in RPMI-1640, D-384 on DMEM and CHO K-1 on HAM F-12 medium supplemented with 10% fetal bovine serum (FBS, Invitrogen, Karlsruhe, Germany), 1% antibiotic-antimitotic solution (100 units/mL penicillin G, 100 µg/mL streptomycin, and 0.25 µg/mL amphotericin B, Gibco, Grand Island, NY, USA), and 1% L-glutamine (2 mM, Gibco). The cells were incubated at 37 °C in a 5% CO_2_ atmosphere. The viable cells were counted using the trypan blue exclusion method in a hemocytometer [62].

#### 4.2.4. Cell Viability Analysis and Determination of the Inhibitory Concentration 50 (IC_50_) on Mammalian Cancer Cells Lines

The MTS (5-[3-(carboxymethoxy)phenyl]-3-(4,5-dimethyl-2-thiazolyl)-2-(4-sulfo- phenyl)-2*H*-tetrazolium inner salt) cell viability assay was used to assess the inhibitory effects of the extracts on the survival of human cancer cell lines. A total of 3–5 × 10^3^ cells/well were seeded into 96-well plates and were allowed to adhere for 24 h. The cells were then treated with 50 µg/mL of whole extract to yield a final volume of 2 mL. Each concentration/assay was performed three times. Dimethyl sulfoxide (DMSO) was used as a negative control at a final concentration of 0.1% *v*/*v*, and 1 µM Doxorubicin was used as a positive control. The cells were incubated with the treatments for 48 h, after which 20 µL MTS (5 mg/mL, Aqueous One Solution Reagent, Gibco) was added and further incubated for 4 h at 37 °C. The absorbance was measured at 570 nm. The data obtained with cells treated with DMSO were defined to represent 100% viability. The IC_50_ was calculated in cell lines with an inhibition percentage over 50% for which five different doses were applied (15, 45, 60, 75, 100, 125 and 150 µM) using nonlinear regression [62].

#### 4.2.5. Cell Viability Analysis by Fluorescein Diacetate/Ethidium Bromide

A total of 8 × 10^3^ CHO-K1 cells were seeded in each well. After 24 h, cells were treated with derivatives in a concentration of 100 µg/mL. Doxorubicin (2 µM) was used as positive control and DMSO (0.25%) as the negative control. Cells were incubated for 24 h in a humidified incubator (37 °C, 5% CO_2_). A cellular suspension was obtained after treatment. This was centrifuged and decanted. The cells were maintained at 4 °C. Cell viability was determined using fluorescein diacetate-ethidium bromide. Cells were stained in 20 µL of a solution of Fluorescein diacetate (5 mg/mL) (FDA, Sigma Aldrich, Saint Louis, MO, USA)/Ethidium Bromide (0.2 mg/mL) (EtBr, Promega, Madison, WI, USA) and observed under a fluorescent microscope (ZEISS-Axioskop 2 plus, filter No. 4, objective 40×, Carl Zeiss, Thornwood, NY, USA). Living cells were stained in green, while dead cells exhibited red nuclei. The survival percentage was obtained by dividing the number of living cells by the total number of cells. All experiments were performed in duplicate using 200 cells per slide.

#### 4.2.6. Comet Assay

Cells were mixed with 150 μL low-melting-point agarose 1% (LMP, Promega, Madison, WI, USA) and added to microscope slides previously prepared with normal-melting-point agarose 1% (NMP, Invitrogen, Carlsbad, CA, USA). A third LMP agarose layer (150 µL) was added. The slides were immersed in a cooled lysing solution (pH 10) with 10% DMSO, 1% Triton X-100 (Sigma Aldrich, Saint Louis, MO, USA), 2.5 M NaCl (Loba Chemie, Mumbai, India), 100 mM EDTA and 10 mM Tris (Invitrogen, Carlsbad, CA, USA) for 12 h at 4 °C. All steps after lysis were performed under darkness or yellow light to prevent additional DNA damage. The slides were immersed for 20 min in an electrophoresis buffer solution (pH 13) with NaOH 300 mM (Fisher Scientific, Waltham, MA, USA) and EDTA 1 mM (Invitrogen, Carlsbad, CA, USA) to allow for DNA unwinding. Electrophoresis was later performed at 25 V and 300 mA for 20 min. Slides were then sprayed with a neutralization buffer (pH 7.5) 0.4 M Tris (Sigma Aldrich, Saint Louis, MO, USA). Ethidium bromide (60 µL) was added to each slide, and a cover glass was placed on the gel. The DNA migration was analyzed using a ZEISS-Axioskop 2 plus microscope with fluorescence (40×) by scoring 100 cells per slide in duplicate. After that, the tail moment parameter was analyzed by a Comet Assay IV software (Perceptive Instruments Ltd., Bury St Edmunds, Suffolk, UK).

For leishmania, a method previously described for yeast was adapted [63]. Briefly, after incubating the cells in presence of the compounds for 24 h, cells were collected and washed with buffer S (1 M sorbitol, 25 mM KH_2_PO_4_ in ultrapure water and adjust to pH 6.5 with NaOH). The pellet was resuspended in 400 µL of 1.5% LMA and 40 µL was spread (second layer) onto previously agarose-coated slides (first layer).

Slides were immersed in cold lysis buffer (30 mM NaOH, 1 M NaCl, 0.05 % (*w*/*v*) lauroylsarcosine, 50 mM EDTA, and 10 mM Tris–HCl, pH 10) for 25 min and then in cold electrophoresis alkaline buffer (30 mM NaOH, 10 mM EDTA, and 10 mM Tris–HCl, pH 10). They were then subjected to electrophoresis at 26V and 300 mA. Slides were immersed in neutralization buffer (10 mM Tris–HCl, pH 7.4; prepared in advance and stored at 4 °C) for 10 min at room temperature. The cells were then treated with proteinase K (20 min, 37 °C in humid chamber), followed by a new neutralization step. To dehydrate the slides, they were immersed in EtOH 90% and EtOH absolute for 10 min each. Finally, the slides were stained with Sybr Gold 1X.

The DNA migration was analyzed using a LEICA microscope with fluorescence (40×) by scoring 100 cells per slide in duplicate. The tail moment parameter was then analyzed by the Comet Score software.

#### 4.2.7. Statistical Analyses

Data were plotted with the statistical software GraphPad Prism 7.02 (GraphPad Software, Corp., San Diego, CA, USA). Kruskal–Wallis test and Tukey’s post-hoc test were run by using agricolae pack-age in R environment.

#### 4.2.8. DPPH Radical Scavenging Assay

The stock solution of the compounds was prepared by dissolving **4** in dimethyl sulfoxide (DMSO) to a concentration of 4 mg/mL. The solution was diluted with methanol until a concentration of 400 µg/mL was obtained and then used immediately.

The experimental procedure was adapted from the literature [64]. Briefly, 100 µL of a 0.2 mM methanol solution of DPPH (2, 2-diphenyl-1-picrylhydrazyl) radical were added to 100 µL of methanolic solutions of **4** prepared as serial two-fold dilutions from the stock solution in 96-well microfilter plates. Standards and edaravone were also prepared in the same concentrations. The mixture was incubated in dark at room temperature for 30 min and the absorbance was read at 515 nm on a Cytation 5 (BioTek) spectrophotometer.

The % DPPH scavenging activity was then calculated by using the following formula:(1)$$\%\;\mathrm{DPPH}\;\mathrm{scavenging}\;=\;100\;\times \;\left[\frac{\left(A_{\mathrm{sample}+\mathrm{DPPH}}-A_{\mathrm{sample}\;\mathrm{blank}}\right)}{\left(A_{\mathrm{DPPH}}-A_{\mathrm{solvent}}\right)}\right]$$

The antioxidant activity of the compound was expressed as IC_50_, which is defined as the concentration that could scavenge 50% of the DPPH free radical. The IC_50_ values were calculated in GraphPad Prism 7.02 (GraphPad Software, Corp.) The results are given as a mean ± standard deviation (SD) of experiments done in triplicate.

#### 4.2.9. Evaluation of Leishmanicidal Activity

The leishmanicidal activity of all compounds was evaluated by measuring promastigotes’ mitochondrial activity using MTT colorimetric assay as described previously [65]. In this study, promastigotes from *Leishmania mexicana* were used. The species were confirmed by amplifying and sequencing cytochrome b gene [66]. Promastigotes were cultured at 25 °C in Schneider’s Drosophila Medium (Gibco, Invitrogen, Carlsbad, CA, USA) supplemented with 10% fetal bovine serum (Eurobio, Les Ulis, France). The medium was renewed every third day. Parasite density was determined by counting the cells using a Neubauer chamber.

Into each well of a 96-well plate were dispensed 1 × 10^6^ parasites/well. A stock solution of the compounds was prepared in DMSO (Sigma Aldrich, Saint Louis, MO, USA), and serial dilutions were added to the parasite suspension, leading to concentrations ranging from 100 to 0.001 µM, keeping the solvent concentration at 0.5%. The final volume was 200 µL for each well and triplicate conditions were carried out. Amphotericin B (Gibco) treatment (1 µM) untreated parasites and DMSO 0.5% were used as positive and negative controls, respectively. After exposure to the compounds for 48 h in culture medium, 20 µL of a solution of 5 mg/mL MTT dissolved in PBS were added to each well. The plate was incubated at 25 °C for 2 h in darkness. The plate was then centrifuged at 4400 rpm for 10 min and the culture medium was then aspirated; 50 µL of DMSO were added into each well to solubilize the formazan crystals and the plate was shaken for 5 min and then it was measured by recording changes in absorbance at 570 nm using a microplate reader Cytation 5 (BioTek) spectrophotometer. A reference wavelength of 630 nm was used as background subtraction. Optical densities were analyzed because the quantity of formazan is directly proportional to the number of viable parasites. Data were analyzed with the statistical software GraphPad Prism 7.02 (GraphPad Software, Corp.).

#### 4.2.10. Evaluation of the Cell Viability

Raw 264.7 (ATCC^®^ TIB-71™) was maintained in Dulbecco’s Modified Eagle Medium (DMEM) (Gibco, Invitrogen, Gibco, Carlsbad, CA, USA) supplemented with 10% fetal bovine serum (FBS) (Eurobio, Les Ulis, France) and 100 IU/mL penicillin + 100 μg/mL streptomycin (Gibco), at 37 °C in a 5% CO_2_ atmosphere. The medium was renewed once a week. The viability was determined using MTT (Thiazolyl Blue Tetrazolium Bromide) dye assay as described before for leishmanicidal activity assessment, with some variations. Here, 5 × 10^4^ cells/well in a final volume of 100 µL were deposited into a 96-well plate, in triplicate. Saponin (2.4 mg/mL) and untreated cells were used as positive control and negative control, respectively. A stock solution was prepared in DMSO and immediately diluted in media to obtain different serial concentrations (100–0.01 µM). After 48 h exposure to the compounds, 10 µL/well of MTT (5 mg/mL MTT in PBS) were added, and the plate was incubated at 37 °C for 2 h in the dark. Cells were pelleted by centrifugation at 4400 rpm for 10 min and the media was removed. Then, 100 µL/well of DMSO were added and the absorbance at 570 nm was recorded; a reference wavelength of 630 nm was used for background subtraction.

## 5. Conclusions

In summary, our results indicate that leishmania are sensitive to most of the bis(spiropyrazolone)cyclopropanes species reported, but no major effect is observed in yeast or human cancer cell lines. Our results suggest the good activity of bis(spiropyrazolone)cyclopropanes **4** against *L. mexicana* with **4r-s** being the most promising. These two compounds show a high leishmanicidal activity (IC_50_ 0.19 µM) with reasonable SI ratios (34.2 and 45.3, respectively). They have fewer genotoxic effect in vitro when exposed to mammalian cell lines. At least some of the compounds studied, but not all, can exert activity by inducing DNA damage. Our data do not allow us to distinguish if DNA damage is exerted directly by the bis(spiropyrazolone)cyclopropanes **4**, or indirectly through inhibition of an enzymatic activity.

This work constitutes the first in vitro screening and unveils compounds with a potential application or, on the contrary, little or no promise. However, additional studies are needed to uncover the real uses and applications of these compounds in therapy before a drug can reach the clinic.

## Data Availability

The data presented in this study are openly available in https://doi.org/10.5281/zenodo.5126826 (accessed on 4 August 2021).

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
