# Peer review of "Synthesis and Evaluation of Biological Activities of Bis(spiropyrazolone)cyclopropanes: A Potential Application against Leishmaniasis"

_molecules, 2021, doi:10.3390/molecules26164960_

Round 1

Reviewer 1 Report

The chemistry reported is a repeat of the author's previously published work, in which compounds 2 were described.  The only new compounds reported in this manuscript in this series are 2f-i,m.  The electrochemical formation of compounds 4 from 2 also has literature precedent; however, the authors should provide more details of the experimental set-up: electrode sizes. In addition, the authors describe these compounds as unstable "even at low temperature" (line 125).  The authors must quantify this statement with, at the least, approximate half-lives at room temperature and at the temperature of the assays.  With out this information, is it not at all clear what the assay results with these compounds represent - that is, how do the authors know that the compounds did not decompose before/during these assays?

The presentation of biological data for these compounds should be modified.  In Table 2, the authors report IC50 values for compounds 4 against cancer cell lines.  Data for many of the compounds is missing, as these were deemed not "showing previously significant cellular inhibition" based on a 100 µM MTT assay.  However, compound 4e, listed as "not evaluated based on initial non-inhibition" had nearly identical activity in HeLa cells when compare to compound 4d, which was evaluated.  Thus, the exclusion of certain compound from Table 4 seems arbitrary.   The results from S. cerevisiae, also listed in Table 4, and for mutant strains in Figure 1, are not informative, as none of these compounds appear to be permeable to these cells.  If the authors want to include these data, they should be reported in the SI, not the main body of the manuscript.  Finally, and not withstanding the above-mentioned concerns over the data for compounds 4 due to instability, the data presented in Table 3 for the Cyctotoxic and Leishmanicidal activity is confusing.  It is not clear how the statistical analyses were carried out - what is being tested / the null hypothesis here?  The authors should clearly state what is being compared here. Similarly for the statistics for Comet assay also need additional explanation.

Author Response

Response to Reviewer 1 Comments

We are grateful to Reviewer 1 for their critical comments that help us to improve our manuscript.

Point 1. The chemistry reported is a repeat of the author's previously published work, in which compounds 2 were described.  The only new compounds reported in this manuscript in this series are 2f-i,m.  The electrochemical formation of compounds 4 from 2 also has literature precedent; however, the authors should provide more details of the experimental set-up: electrode sizes.

Response 1. The procedure for the electrochemical reaction was updated with more details in the Materials and Methods section (lines 450-456).

Point 2. In addition, the authors describe these compounds as unstable "even at low temperature" (line 125).  The authors must quantify this statement with, at the least, approximate half-lives at room temperature and at the temperature of the assays.  With out this information, is it not at all clear what the assay results with these compounds represent - that is, how do the authors know that the compounds did not decompose before/during these assays?

Response 2. We provide data concerning the stability of compounds. Data has been added in the Supporting Material (Figure S3). First, we previously tested the Inhibitory activity of products of thermal isomerization of bis(spiropyrazolone)cyclopropanes 4 against L. mexicana promastigotes at 10 μM (Figure S3 D), and none of the compounds assayed reach the 50% inhibition. Conversely, the non-thermal isomerized bis(spiropyrazolone)cyclopropanes 4 inhibited the 50 % of population at concentrations between 2.23 and 0.15 µM (Table 3). In our opinion these data provide reliable arguments supporting the leishmanicidal activity of bis(spiropyrazolone)cyclopropanes 4. Second, we have now calculated the stability of compound 4b in aqueous solution (PBS) and in DMSO at 25°C, ie the temperature for leishmania culturing. To perform these analyses, we chose compound 4b since it is one of the most active against leishmania, and has not substituents on the ring. Results are shown in Figure S3 (A-C). The color change due to the thermal isomerization of compound 4b in DMSO after 24 h at 25°C is easily observed visually. 

This information has been added in the main text (lines 202-211).

Point 3. The presentation of biological data for these compounds should be modified.  In Table 2, the authors report IC50 values for compounds 4 against cancer cell lines.  Data for many of the compounds is missing, as these were deemed not "showing previously significant cellular inhibition" based on a 100 µM MTT assay.  However, compound 4e, listed as "not evaluated based on initial non-inhibition" had nearly identical activity in HeLa cells when compare to compound 4d, which was evaluated.  Thus, the exclusion of certain compound from Table 4 seems arbitrary.  

Response 3. The following criteria were applied for the selection of the compounds to evaluate the IC50 in the cell lines: i) We selected the most sensitive lines, in this case: RKO, PC-3 and Hela (Table S1). Ii) The compounds had to inhibit more than 50%, in at least one cell line and we established the IC50 in the 3 most sensitive lines.

In the case of compound 4e, it only has inhibition values similar to that of 4d for HeLa, but it does not inhibit any cell line by more than 50%, therefore, the determination of the IC50 was not continued.

These criteria have been added in the manuscript for better understanding (lines 135-144).

Point 4. The results from S. cerevisiae, also listed in Table 4, and for mutant strains in Figure 1, are not informative, as none of these compounds appear to be permeable to these cells.  If the authors want to include these data, they should be reported in the SI, not the main body of the manuscript. 

Response 4. The suggestion is accepted and results have been reported to the Supporting Information. Inhibitory activity on S. cerevisiae is now renamed as Table S2, and Figure 1 as Figure S2.

Point 5. Finally, and not withstanding the above-mentioned concerns over the data for compounds 4 due to instability, the data presented in Table 3 for the Cyctotoxic and Leishmanicidal activity is confusing.  It is not clear how the statistical analyses were carried out - what is being tested / the null hypothesis here?  The authors should clearly state what is being compared here. Similarly for the statistics for Comet assay also need additional explanation.

Response 5. We reformulated the statistical analyses and rewrote Table 3. The level of activity was log-transformed (log(x+1)) in order to reach homoscedasticity criteria, and one-way ANOVA test was applied and for the post-hoc comparison Tukey test was applied. The significance level was set at 5% in the whole study.

Table 3. Inhibitory activity of bis(spiropyrazolone)cyclopropanes 4 against promastigotes of L. mexicana and cytotoxicity in RAW cells and selective indexes.

Leishmanicidal activity

IC50 µMa

Cytotoxicity

CC50 µM

SIb

4b

0.16 d

3.44 cd

21.3

4c

0.15 d

2.28 d

15.3

4d

0.15 d

3.92 bcd

26.0

4e

1.48 b

7.28 abc

4.9

4f

2.23 a

12.82 a

5.7

4k

0.30 cd

3.31 cd

11.0

4p

0.57 c

3.55 cd

6.1

4q

0.28 cd

4.00 bcd

14.3

4r

0.19 d

6.50 abc

34.2

4s

0.19 d

8.63 ab

45.3

4t

0.22 cd

2.75 d

12.3

4u

0.42 cd

9.30 a

22.1

Ampho B

0.17 d

>31.8

a The tests for significance were based on one-way ANOVA test. For this, the level of activity was log-transformed (log(x+1)) in order to reach homoscedasticity criteria, and for the post-hoc comparison Tukey test was applied. The level of significance was set at 5%. Treatments with the same letter are not significantly different.; b SI (Selectivity Index) = Citotoxic activity / Leishmanicidal activity.

We have also updated information from Figures 2 and 3 (lines 247-251; 287-291). Details of the statistical analyses are in Supplementary material.

Reviewer 2 Report

The manuscript molecules-1302950 "Synthesis and evaluation of biological activities of bis(spiropyrazolone)cyclopropanes: a potential application against leishmaniasis" by Barreiro et. al. describes the synthesis of 12 derivatives of bis(spiropyrazolone)cyclopropanes and the study of their biological activities. Synthesis of new compounds was confirmed by series of physical methods (1H, 13C NMR spectroscopy, mass-spectrometry). As a result, two derivatives of bis(spiropyrazolone)cyclopropanes are good candidates for the treatment of leishmaniasis and have specificity against parasites with respect to mammalian cells.

Questions and comments:

1) The synthesis of heterocyclic compounds including derivatives of cyclopropanes is a well-researched topic. Unfortunately, the introduction contains very low references to studies of the last 5-10 years. I recommend that authors add references to more recent research to understand the relevance of the topic.

2) How was the stereochemical configuration of asymmetric carbon atoms confirmed in the obtained compounds?

3) Table 2 lacks doxycycline control activity data.

4) I recommend to add a separate Conclusions section in which the results obtained will be summarized.

5) Lines 296-394. "Discussion" part contains text similar to "Introduction" part. Information about the relationship of the chemical structure of the obtained compounds with biological activity, especially the compounds with the highest activity, should be added.

Author Response

Response to Reviewer 2 Comments

We are also grateful to Reviewer 2 for their critical comments that help us to improve our manuscript.

The manuscript molecules-1302950 "Synthesis and evaluation of biological activities of bis(spiropyrazolone)cyclopropanes: a potential application against leishmaniasis" by Barreiro et. al. describes the synthesis of 12 derivatives of bis(spiropyrazolone)cyclopropanes and the study of their biological activities. Synthesis of new compounds was confirmed by series of physical methods (1H, 13C NMR spectroscopy, mass-spectrometry). As a result, two derivatives of bis(spiropyrazolone)cyclopropanes are good candidates for the treatment of leishmaniasis and have specificity against parasites with respect to mammalian cells.

Questions and comments:

Point 1. The synthesis of heterocyclic compounds including derivatives of cyclopropanes is a well-researched topic. Unfortunately, the introduction contains very low references to studies of the last 5-10 years. I recommend that authors add references to more recent research to understand the relevance of the topic.

Response 1. We update the references in the introduction to cover more recent research about cyclopropanes. Thank you very much for the suggestions.

Point 2. How was the stereochemical configuration of asymmetric carbon atoms confirmed in the obtained compounds?

Response 2. In the synthesis of 4, the formation of a pair of diasteromers with (R*,R*) or (R*,S*) relative configuration is possible, but the spectroscopic evidence clearly indicated that the only diasteromer formed is (R*,R*). For example, in the 1H-NMR of 4b, the two methyl protons of the pyrazole rings appeared as two singlets at 2.09 ppm and 2.53 ppm. The molecule of the (R*,R*) compound has the methyl groups of the pyrazolone rings on different sides of the cyclopropane plane, so one of them is cis to the phenyl at C11 while the other one is trans. Both methyl groups are in a different chemical environment, which justifies the different chemical shift observed experimentally.

On the other hand, If the diastereomer (R*,S*) had formed, the methyl groups of the pyrazolone rings would appear as a singlet since they are on the same side of the cyclopropane plane (cis or trans to the aromatic ring at C11), so the chemical environment would be the same for both methyl groups. In the 1H-NMR spectra of all synthesized compounds, no evidence of the presence of the diasteromer (R*,S*) is observed.

Finally, the relative configuration (R*,R*) of these compounds was determined experimentally by Elinson, M. N., et al. Synthesis, 2011, 18, 3015–3019. (https://doi.org/10.1055/s-0030-1261031), and since we used the same procedure reported for the synthesis, the same relative configuration is expected.

Point 3. Table 2 lacks doxycycline control activity data.

Response 3. The data was added (Table 2).

Point 4. I recommend to add a separate Conclusions section in which the results obtained will be summarized.

Response 4. We agree with the suggestion and we wrote the Conclusions section separately (line 394).

Point 5. Lines 296-394. "Discussion" part contains text similar to "Introduction" part. Information about the relationship of the chemical structure of the obtained compounds with biological activity, especially the compounds with the highest activity, should be added.

Response 5. Some discussion about the relationship of the chemical structure of compounds 4 and their leishmanicidal activity was add to the manuscript as the reviewer suggested (lines 74-81; 366-375).

Round 2

Reviewer 1 Report

Please post the revised Supporting Information - I can not evaluate this revision without this.

Reviewer 2 Report

I thank the authors for answering my questions and improving the manuscript.